# Bile Duct Ligation Upregulates Expression and Function of L-Amino Acid Transporter 1 at Blood–Brain Barrier of Rats via Activation of Aryl Hydrocarbon Receptor by Bilirubin

**DOI:** 10.3390/biomedicines9101320

**Published:** 2021-09-26

**Authors:** Xiaoke Zheng, Hanyu Yang, Lan Qin, Siqian Wang, Lei Xie, Lu Yang, Weimin Kong, Liang Zhu, Li Liu, Xiaodong Liu

**Affiliations:** Center of Drug Metabolism and Pharmacokinetics, School of Pharmacy, China Pharmaceutical University, Nanjing 210009, China; zhengxk0923@yeah.net (X.Z.); shenyhycpu@163.com (H.Y.); 17878076997@163.com (L.Q.); siqianwang@yeah.net (S.W.); asdfghjkl951415@163.com (L.X.); yanglu2214@163.com (L.Y.); 1831010057@stu.cpu.edu.cn (W.K.); a8459022@126.com (L.Z.)

**Keywords:** L-amino acid transporter, blood–brain barrier, liver failure, unconjugated bilirubin, aryl hydrocarbon receptor

## Abstract

Liver failure is associated with increased levels of brain aromatic amino acids (AAAs), whose transport across the blood–brain barrier (BBB) is mainly mediated by L-amino acid transporter 1 (LAT1). We aimed to investigate whether liver failure induced by bile duct ligation (BDL) increases levels of brain AAAs by affecting the expression and function of LAT1. The LAT1 function was assessed using the brain distribution of gabapentin. It was found that BDL significantly increased levels of gabapentin, phenylalanine, and tryptophan in the cortex, hippocampus, and striatum of rats, and upregulated the expression of total LAT1 protein in hippocampus and striatum as well as cortex membrane LAT1 protein. HCMEC/D3 served as in vitro BBB model, and the data showed that both the serum of BDL rats and bilirubin induced LAT1 expression and function, while bilirubin oxidase almost abolished the upregulation of LAT1 protein by bilirubin and the serum of BDL rats. The enhanced function and expression of LAT1 were also observed in the hippocampus and striatum of hyperbilirubinemia rats. Both aryl hydrocarbon receptor (AhR) antagonist α-naphthoflavone and AhR silencing obviously attenuated the upregulation of LAT1 protein by bilirubin or omeprazole. This study provides the first evidence that BDL upregulates LAT1 at the rat BBB, attributed to the activation of AhR by the increased plasma bilirubin. The results highlight the mechanisms causing BDL-increased levels of brain AAAs and their physiological significance.

## 1. Introduction

Liver failure is often associated with hepatic encephalopathy, accompanied by a series of neuropsychiatric syndromes such as impairment of cognitive function, alterations in motor activity, coordination or coma [1,2]. It is generally accepted that hepatic encephalopathy results from the accumulation of neurotoxic and/or neuroactive substances in the brain [1,2]. Several studies [3,4,5,6,7,8,9,10,11,12,13] have demonstrated that liver failure significantly elevates concentrations of aromatic amino acids (AAAs) in the brain or cerebrospinal fluid, including phenylalanine (Phe), tyrosine (Tyr), and tryptophan (Trp), whose increases are often positively related to grades of hepatic encephalopathy [3,4,6]. However, the real mechanism by which liver failure increases concentrations of brain AAAs has not been fully investigated.

It is noteworthy that these AAAs are precursors of monoamine neurotransmitters dopamine (DA) and serotonin (5-HT), suggesting that the increases in brain AAAs may influence the synthesis of 5-HT and DA, later precipitating hepatic encephalopathy. Several investigations support the above deduction. For example, significantly increased levels of 5-HT and DA have been observed in cerebrospinal fluids of hepatic encephalopathy patients [5]. The increased levels of 5-HT are also found in most brain regions of hepatic encephalopathy patients [14]. Animal experiments showed that thioacetamide-induced liver failure significantly increases levels of DA in the hippocampus of rats and impairs learning and memory abilities, characterizing minimal hepatic encephalopathy [15]. The increased concentration of cortex DA by intracranial dose also induced similar minimal hepatic encephalopathy [16]. Moreover, thioacetamide-induced liver failure significantly increases levels of 5-HT in the brain of mice and impairs cognitive function [17,18]. Our preliminary results also demonstrated that liver failure induced by bile duct ligation (BDL) increases the concentrations of both DA and 5-HT in the brain of rats, which are in line with the increase in levels of Phe and Trp.

Under physiological conditions, the homeostasis of AAAs in the brain is highly dependent on the blood–brain barrier (BBB) [19]. The transport of AAAs across the BBB is mainly mediated by the large neutral amino acid transporter 1 (LAT1/SLC7A5). LAT1, a sodium and pH-independent transmembrane transporter, shows its transport activity in a heterodimeric complex with glycoprotein CD98 (4F2hc/SLC3A2) [20]. LAT1 is highly expressed in brain microvessel endothelia cells [20,21], where LAT1 transports large neutral amino acids such as Phe, Trp, Tyr, histidine (His), leucine (Leu), valine (Val), methionine (Met), and isoleucine (Ile). In addition, the preferential transport order of LAT1 is Phe > Trp > Leu > Ile > Met > His > Tyr > Val [22]. LAT1 also mediates transport of some drugs such as gabapentin, pregabalin, L-DOPA, and melphalan [23,24,25,26], and facilitates cellular uptake of thyroid hormones [27]. Some amino acid-related neurotoxins, such as methylmercury-cysteine conjugate, beta-N-methylamino-L-alanine, S-(1,2-dichlorovinyl)-L-cysteine, and 3-hydroxykynurenine [28] are also substrates of LAT1. Some diseases such as brain metastases [29,30] and epileptic seizure [31] remarkably increase the function and expression of LAT1 in the brain. The increased LAT1 expression is also detected in the brain of patients with tuberous sclerosis complex [32] and in brain samples of glioblastoma patients [33,34]. These results indicate that liver failure may increase levels of brain Phe and Trp by upregulating the expression and function of brain LAT1.

The study aimed to investigate whether liver failure increases the expression and function of LAT1 at the rat BBB, and in turn, increases levels of brain AAAs. The BDL rat model, characterizing chronic cholestasis, is the most widely used model of type C hepatic encephalopathy [35]. Brain distribution of LAT1 substrate gabapentin [26] was indexed as the function of LAT1. LAT1 is also highly expressed in the human cerebral microvessel endothelial cells (HCMEC/D3) [25,26], thus the cell lines were served as an in vitro BBB model to screen factors upregulating the expression and function of LAT1 under liver failure. BDL rats often characterize hyperbilirubinemia. The final aim was to further verify the roles of bilirubin in the upregulation of LAT1 expression and function in the BBB of BDL rats using hyperbilirubinemia rats (HB). The results highlight alterations in LAT1 function and expression in the BBB caused by liver failure and their physiological significances.

## 2. Materials and Methods

### 2.1. Reagents

Gabapentin, ammonium chloride and unconjugated bilirubin (UCB) were purchased from Sigma-Aldrich (St. Louis, MO, USA). Omeprazole, α-naphthoflavone, and the bilirubin oxidase (BOX) were obtained from Aladdin Co., Ltd. (Shanghai, China). Phenylalanine (Phe) and tryptophan (Trp) were from Macklin Biochemical Co., Ltd. (Shanghai, China). Fetal bovine serum and RPMI-1640 medium were from ScienCell Research Laboratories (San Diego, CA, USA) and Invitrogen Co (Carlsbad, CA, USA), respectively. Bicinchoninic acid (BCA) kit, 5× loading buffer, and RIPA lysis buffer were from Beyotime Biotechnology (Nanjing, China). The membrane and cytoplasmic protein extraction kits were from KeyGEN BioTECH (Nanjing, China). Analysis kits for serum ammonia, total bile acids, total bilirubin, alanine aminotransferase (ALT), aspartate aminotransferase (AST), and alkaline phosphatase (ALP) were from Jiancheng Bioengineering Institute (Nanjing, China). LAT1 (sc-374232) antibody was from Santa Cruz Biotechnology (Dallas, TX, USA). CD98 polyclonal antibody was purchased from ImmunoWay Biotechnology (Plano, TX, USA). β-actin antibody was obtained from Bioworld Technology (Nanjing, China). CD71 antibody and aryl hydrocarbon receptor (AhR) antibody were from Cell Signaling Technology (Danvers, MA, USA) and Wanleibio (Shenyang, China), respectively. All the other reagents employed were of analytical grade and were commercially available.

### 2.2. Animals

Male Sprague-Dawley (SD) rats, weighing 220–250 g, were purchased from Sin-British Sippr/BK Laboratory Animal Ltd. (Shanghai, China). They were free to have commercially available food and water, under controlled environmental conditions (temperature, 24 ± 2 °C; humidity, 50 ± 5%, 12 h light/dark cycle) during the 7 day acclimation. The animal experiments were carried out by the guidelines on the Care and Use of Animal developed by the National Advisory Committee for Laboratory Animal Research in accordance with ARRIVE guidelines and were approved by the Ethics Committee of the Animal Care Council of the China Pharmaceutical University (protocol code 201902005 and the data of approval 15 February 2019).

### 2.3. Development of BDL Rats

BDL rats were developed according to the method previously described in [36]. Sham-operated (sham) rats received the same experimental procedure without bile duct ligation. On day 14 following the operation, 8 BDL rats and sham rats were used for the following experiments.

### 2.4. Brain Distributions of Gabapentin and AAAs in Rats

On the 14th day following BDL, concentrations of gabapentin, Phe, and Trp in the brain of experimental rats were detected. Briefly, the experimental rats, fasted overnight, were sacrificed under light diethyl ether at 40 min after intravenous injection of gabapentin (15 mg/kg). The spleens and livers were weighed and serum samples were immediately collected to measure biochemical parameters. Brain tissues (cerebral cortex, striatum, and hippocampus) were collected for detecting levels of gabapentin, Phe, and Trp as well as expression of LAT1 and CD98 protein. Plasma samples were also obtained for assessing levels of gabapentin, Phe, and Trp.

The concentrations of gabapentin in plasma and brain tissues, following precolumn derivatization, were quantitated by HPLC-fluorimetric detection [37]. The calibration curves in plasma and brain homogenate were linear over the concentration ranges of 0.0625–20 μg/mL. The levels of Phe and Trp in brain tissues and plasma were detected by liquid chromatography–mass spectrometry (LC–MS). The calibration curves in plasma and brain homogenate were linear over the concentration ranges of 0.625–40 μg/mL and 0.039–5 μg/mL for Phe and Trp, respectively. Serum biochemical parameters including total bilirubin, serum ammonia, total bile acid levels, and activities of ALT, AST, and ALP were determined by commercial reagent kits according to their instructions.

### 2.5. HCMEC/D3 Culture and Drug Treatment

HCMEC/D3 was provided by JENNIO Biological Technology Ltd. (Guangzhou, China). HCMEC/D3 were seeded in 6-well plates at a density of 2 × 10^5^/well in RPMI-1640 medium containing 10% fetal bovine serum. When confluence was reached (about 70–80%), the cells were cultured in RPMI-1640 medium containing 25% serum of experimental rats, UCB (10, 25, and 50 μM) and NH_4_Cl (0.2, 1, and 5 mM) for 24 h, respectively. The concentrations of UCB and NH_4_Cl we used in this study were taken from previous reports [38,39,40]. The cultured cells were collected for determining the protein expression and function of LAT1. The MTT assay showed that these agents did not damage cell viability at the concentrations tested. To further investigate whether serum of BDL rats induced the expression of LAT1 protein due to the increased bilirubin levels, LAT1 protein levels were also measured in HCMEC/D3 cells following treatment with UCB (50 μM) or serum of BDL rats in the absence and presence of bilirubin oxidase (50 μg/mL). The levels of bilirubin oxidase we used were taken from previous reports [41].

### 2.6. Uptakes of Gabapentin in HCMEC/D3

The uptake of gabapentin by HCMEC/D3 was performed to assess the function of LAT1. The concentration of gabapentin in the uptake system and the incubation time were designed based on pre-experimental data (Appendix A) and previous reports [26]. Briefly, the cultured cells were preincubated at 37 °C with a 1 mL Hanks’ balanced salt solution (HBSS) for 15 min. The medium was replaced using 1 mL HBSS containing 100 μM gabapentin to start the uptake reaction. Following a 30 min incubation, the uptake reaction was stopped by ice-cold HBSS. Following 3 times washes using ice-cold HBSS, 0.4 mL of pure water was added to the cells, which was then frozen and melted repeatedly to break down the cells. The amount of gabapentin in cell samples was determined by HPLC. The concentrations of total cell protein were measured by the BCA protein assay kit. Cellular uptake (mL/mg protein) was obtained by dividing the amount of gabapentin in cell samples (μg/mL) by the concentration of cell protein (μg/mg protein).

### 2.7. AhR Knockdown with siRNA in HCMEC/D3 Cells

To estimate the contributions of AhR to the UCB-mediated upregulation of LAT1 expression, AhR knockdown was achieved in HCMEC/D3 cells using AhR siRNA (sense 5′-UCAUGCAGCUGAUAUGCUUTT-3′, antisense 5′-AAGCAUAUCAGCUGCAUGATT-3′) from Gene Pharma Technology (Shanghai, China) and Lipofectamine 3000 according to the manufacturer’s instructions. Briefly, the cells, following 24 h transfection, were cultured in the medium containing UCB (50 μM) or OM (100 μM) for another 24 h, then the protein expression of LAT1 in HCMEC/D3 cells were detected.

### 2.8. Effects of Hyperbilirubinemia on LAT1 Function and Expression in the Brain of Rats

To further study the effects of UCB on the function and expression of LAT1 in the BBB, hyperbilirubinemia rats were induced by intraperitoneal dose of UCB (85.5 μmol/kg, once) for 7 days (HB-7d) and 14 days (HB-14d), according to the method previously described [37,42]. The control rats only received the vehicle; every group had 5 rats. On the experimental day, at 10 min following the last dose of UCB, the experimental rats intravenously received gabapentin (15 mg/kg). Then, the rats were sacrificed under light diethyl ether at 40 min following dose of gabapentin. Plasma and brain samples were obtained, then the levels of gabapentin, Phe, and Trp in the brain and plasma as well as expression of LAT1 in the brain were measured by using the same methods described above.

### 2.9. Western Blot

The protein levels of LAT1 in brain tissues (cortex, hippocampus, and striatum) and HCMEC/D3 cells were measured by Western blot. Briefly, brain tissues and cell samples were homogenized in a RIPA lysis buffer. The plasma membrane of the cerebral cortex was also prepared using a membrane protein extraction kit (Keygen Biotech Corp, Nanjing, China) to detect the expression of membrane LAT1 protein. The amounts of proteins were determined by using a BCA protein assay kit. Equal amounts of proteins were separated by sodium dodecyl sulfate─polyacrylamide gel electrophoresis and transferred to nitrocellulose membranes (Millipore, Billerica, MA, USA). After being blocked with 5% nonfat dry milk solution dissolved in Tris-buffered saline containing 0.1% Tween 20 at room temperature for 2 h and then washed, the membranes were incubated with rat monoclonal anti-LAT1 (1:200 dilution), anti-β-actin (1:8000 dilution), anti-CD71 (1:600 dilution), anti-AhR (1:1500 dilution), and anti-CD98 (1:1000) overnight at 4 °C. The membranes were washed three times by Tris-buffered saline containing 0.1% Tween 20 and then incubated in HRP-conjugated secondary anti-rabbit antibody (1:3000 dilution) for 1.5 h or anti-rat antibody (1:3000 dilution) for 1.5 h. The immunoreactivity was detected by SuperSignal West Femto Chemiluminescent Substrate (Thermo Fisher Scientific Inc., Waltham, MA, USA) and a gel imaging system (ChemiScope 2850, Tanon Technology, Co., Ltd., Shanghai, China). The intensity values were normalized to the intensity of β-actin for total proteins and to the intensity of CD71 for membrane proteins.

### 2.10. Statistical Analysis

All data are presented as mean ± standard derivation (SD). *t*-tests were used for assessing the two groups and a one-way ANOVA followed by least significant difference tests were used to assess the comparisons among multiple groups. A *p* < 0.05 was regarded as indicating statistical significance.

## 3. Results

### 3.1. Biochemical and Physiological Parameters in BDL Rats

BDL-induced liver failure was confirmed by the evaluation of physiological and biochemical parameters (Table 1). As expected, the levels of ALT, AST, ALP, total bile acid, total bilirubin, and ammonia in the serum of BDL rats were significantly higher than those in the serum of sham rats, indicating that BDL-induced liver failure rats were successfully developed.

### 3.2. BDL Increases the Function and Expression of LAT1 in Brain Tissues of Rats

The concentrations of gabapentin in brain tissues (cortex, hippocampus, and striatum) and plasma were measured to assess the function of LAT1. The results showed that BDL significantly increased the levels of gabapentin in the cortex, hippocampus, and striatum of rats without affecting plasma concentration of gabapentin (Figure 1A); as a result, the brain-to-plasma concentration ratio of gabapentin was significantly upregulated by BDL (Figure 1B). Our previous report showed that BDL has little effect on the integrity of the BBB [36], suggesting that the increased distribution of gabapentin in the brain of BDL rats is attributed to the enhanced function of LAT1.

The protein levels of LAT1 in the cortex, hippocampus, and striatum of BDL rats were further detected using western blotting. It was found that the BDL remarkably increased the levels of LAT1 protein in the hippocampus and striatum of rats. However, significantly decreased expression of total LAT1 protein was observed in the cortex of BDL rats (Figure 1C,D), which is not consistent with increased levels of gabapentin in the cerebral cortex. Thus, expression of membrane LAT1 protein and its partner CD98 protein in the cerebral cortex of BDL rats were also measured. As expected, the expression of membrane LAT1 protein in the cerebral cortex of BDL rats was significantly higher than that in sham rats (Figure 1E,F). BDL also increased CD98 protein expression in the cortex of rats, although no significance was observed (Figure 1E,F).

The concentrations of Trp and Phe in brain tissues of BDL rats were also measured and the brain-to-plasma concentration ratios were calculated. In line with the increased function of brain LAT1, the BDL significantly increased the levels of Trp and Phe in the cortex, hippocampus, and striatum of rats (Appendix A), and obviously elevated the brain-to-plasma concentration ratio of Trp in the cortex and hippocampus of rats (Figure 1G).

### 3.3. Effects of UCB and Ammonia on Expression and Functions of LAT1 in HCMEC/D3 Cells

HCMEC/D3 cells served as an in vitro model of the BBB to identify which abnormally altered constituents induced expression of LAT1 protein. Data from Western blotting showed that incubation with the serum of BDL rats significantly increased the expression of LAT1 protein (Figure 2A), which was consistent with the in vivo data, indicating that abnormally altered constituents existed in the serum of BDL rats, which induced the expression and function of LAT1 at the BBB. Significantly increased levels of bilirubin and ammonia were observed in the serum of BDL rats (Figure 2B,C). The effects of UCB and ammonia on LAT1 were further investigated. The results showed that only UCB concentration dependently increased the expression and function of LAT1 protein (Figure 2D,E). At a concentration of 50 μM, UCB significantly increased the protein expression and function (gabapentin uptake) of LAT1 to 2.3 times and 1.7 times those of control cells (Figure 2D,E), respectively. At a concentration of 5 mM, NH_4_Cl only showed little enhancement on the uptake of gabapentin (Figure 2F,G). The MTT assay showed that these agents at the concentrations tested did not damage cell viability (Appendix A). Further study showed that the treatment of bilirubin oxidase almost abolished the upregulation of LAT1 protein by UCB and serum of rats (Figure 2H,I), demonstrating that the upregulation of LAT1 protein by serum of BDL rats was mainly attributed to the increased bilirubin levels.

### 3.4. Implications of AhR Pathway in Upregulation of LAT1 Expression by UCB in HCMEC/D3 Cells

Several reports have shown that UCB induces the expression of UGT1A1 [42] and CYP1a1 [43] by activating the AhR pathway. To investigate whether UCB also upregulated the expression of LAT1 by activating the AhR pathway, the expression of LAT1 protein in HCMEC/D3 cells following treatment of UCB, AhR antagonist α-naphthoflavone and AhR silencing were measured. AhR agonist omeprazole served as positive control. The results showed that both omeprazole and UCB remarkably upregulated the expression of LAT1 protein in HCMEC/D3 cells (Figure 3A,B), which could be reversed by α-naphthoflavone (Figure 3A,B). These results implied that AhR is implicated in the upregulation of LAT1 by UCB. To further confirm the roles of AhR in the UCB-mediated upregulation of LAT1 expression, the expression of AhR in HCMEC/D3 cells was silenced using siRNA of AhR (siAHR). The expression of AhR protein in HCMEC/D3 cells transfected with siAHR was decreased to 34% of control cells, suggesting that AhR was successfully silenced (Figure 3C). More importantly, AhR silencing significantly attenuated the upregulation of LAT1 protein by both omeprazole and UCB (Figure 3D,E).

### 3.5. Effects of Hyperbilirubinemia on Brain LAT1 Expression and Function

Hyperbilirubinemia rats, which were induced by an intraperitoneal dose of UCB for 7 days (HB-7d) and 14 days (HB-14d), were further used to verify whether the increases in levels of bilirubin contributed to the upregulation of the brain LAT1 function and expression. Significantly increased levels of total bilirubin and conjugated bilirubin were detected in the serum of both HB-7d and HB-14d rats (Figure 4A), implying that hyperbilirubinemia rats were successfully developed. The brain distributions of gabapentin in HB-7d and HB-14d rats were further measured. The results showed that the effects of hyperbilirubinemia on brain distributions of gabapentin were time-dependent. Compared with that of control rats, the brain of HB-14d rats exhibited higher concentrations of gabapentin, and the difference was significant was found in the hippocampus and striatum of HB-14 rats (Appendix A). The brain-to-plasma concentration ratio of gabapentin also showed an upward trend in the hippocampus and striatum of HB-14d rats, although no significance was obtained (Figure 4B). Total LAT1 protein in the hippocampus, striatum, and cortex of hyperbilirubinemia rats was also measured. The results showed that hyperbilirubinemia significantly increased the levels of brain LAT1 in a time- and brain-region-dependent manner. The extent of increased expressions of LAT1 protein in the hippocampus and striatum of HB-14 d rats was larger than that in HB-7d rats (Figure 4C,D). Hyperbilirubinemia also remarkably increased the expression of LAT1 in the cerebral cortex, but no time-dependence was obtained. The protein expression of CD98 and membrane LAT1 in the cerebral cortex was further detected. It was found that the increased expression of membrane LAT1 protein by hyperbilirubinemia in the rat cortex was also time-dependent (Figure 4E,F). However, levels of CD98 in the cortex of HB-7d rats were obviously lowered, whereas the expression of CD98 in the cortex of HB-14d rats was unaltered (Figure 4E,F).

The concentrations of Phe and Trp in the brain of hyperbilirubinemia rats were also measured. The results showed that Phe concentration significantly increased in the cortex of HB-14d rats, while Trp concentration significantly increased in the cortex of HB-7d rats (Appendix A). The brain-to-plasma concentration ratio was also calculated, and the data showed that hyperbilirubinemia significantly increased the brain-to-plasma concentration ratio of Phe and Trp in the cortex and hippocampus of HB-14d rats, which possibly results from UCB-induced LAT1 upregulation (Figure 4G,H).

## 4. Discussion

This study provides the first evidence that BDL-induced liver failure upregulates the expression and function of LAT1 at the rat BBB, which is evidenced by the increases in the distribution of the LAT1 substrate gabapentin. In general, LAT1 plays a crucial role in providing the brain with essential amino acids (including branched-chain amino acids and AAAs) to maintain normal brain functions [20]. It is in line with our findings that significantly elevated concentrations of Phe and Trp have been demonstrated in the brain under liver failure [3,4,5,6,7,8,9,10,11,12,13]. These results indicate that the increased expression and function of LAT1 caused by BDL may be one of the reasons that liver failure increases levels of Phe and Trp.

In the present study, significantly upregulated levels of LAT1 protein were found in the hippocampus and striatum of BDL rats. However, it was in contrast to the increased concentration of gabapentin in the cortex (indicating an upregulation of LAT1 functions) that BDL significantly decreased the expression of total LAT1 protein in the cortex of rats. In general, the functions of LAT1 are often linked to its membrane locations and heterodimeric complex with its partner CD98 [20], so the expression of membrane LAT1 protein and its partner CD98 were also measured. The remarkably increased expression of membrane LAT1 protein and slightly increased expression of CD98 were observed in the cortex of BDL rats, which may explain the increased distribution of gabapentin in the cortex. It is consistent with the upregulation of LAT1 function that higher levels of Phe and Trp were also found in the hippocampus, striatum, and cortex of BDL rats. Phe and Trp are precursors of DA and 5-HT, respectively. The increased levels of brain Phe and Trp may, at least partly, contribute to the increased levels of DA and 5-HT [unpublished data]. Several reports have demonstrated that DA and 5-HT are implicated in the development of hepatic encephalopathy [14,15,16]. All these results suggest that an enhancement of LAT1 function in the BBB possibly contributes to hepatic encephalopathy.

Next, we investigated the mechanism by which the BDL enhances the functions and expression of LAT1 at the BBB by using HCMEC/D3 as an in vitro BBB model. The serum of BDL rats also increased the LAT1 expression in HCMEC/D3 cells, demonstrating that abnormally altered compounds in the serum of BDL rats may induce expression of LAT1 protein. It is generally accepted that ammonia and bilirubin are the main abnormally elevated components in the plasma of BDL rats, and thus their effects on the expression and function of LAT1 were tested. The results showed that only bilirubin concentration dependently enhanced the function and expression of LAT1 in HCMEC/D3 cells, and BOX treatment obviously decreased the LAT1-inducing effects of bilirubin and serum of BDL rats. The above results demonstrated that increased bilirubin levels in the serum of BDL rats are the main compound inducing LAT1 expression in the BBB.

Bilirubin is considered to be an endogenous ligand of AhR [44,45], and several studies have demonstrated that bilirubin induces the expression of UGT1A1 [42] and CYP1a1 [43] by activating the AhR pathway. Mouse experiment also showed that UCB has immunomodulatory effects, which was also mediated by an AhR-dependent pathway [46]. Activation of AhR has also been demonstrated to upregulate the expression of LAT1 protein in MCF-7, MDA-MB-23 cells [47], and in human bronchial epithelial cells [48]. We found that omeprazole, an AhR agonist, could also induce the expression of LAT1 protein in HCMEC/D3 cells, demonstrating that LAT1 may be a target of AhR. Furthermore, the treatment of AhR antagonist α-naphthoflavone or siRNA of AhR remarkably reversed the upregulation of LAT1 expression by omeprazole or bilirubin. All these results support the conclusion that bilirubin upregulates the expression of LAT1 protein by activating the AhR pathway.

The importance of bilirubin in the expression and function of LAT1 at the BBB by BDL was further confirmed using hyperbilirubinemia rats. It is in line with in vitro findings that hyperbilirubinemia significantly induces the expression of LAT1 in the cortex, hippocampus, and striatum of rats. Time-dependent inductions of LAT1 protein in the striatum and hippocampus were observed. It is in contrast with our expectation that, although the enhanced expression of total LAT1 protein was also found in the hippocampus and striatum of HB-7d rats, the functions of LAT1 were little affected. Similarly, a significantly increased expression of total LAT1 protein and membrane LAT1 protein was also observed in the cortex of HB-7d and HB-14d rats, but the brain distribution of gabapentin was unaltered. The expression of CD98 protein in the cortex was significantly downregulated in HB-7d rats and unaltered in HB-14d rats, which seemed to explain the findings that the expression of LAT1 protein does not match with its function in the cortex of hyperbilirubinemia rats. Moreover, LAT2 also mediates transport of Phe and Trp [49], and LAT1 and LAT2 functions are complementary [50]. Whether hyperbilirubinemia and BDL also alter the expression and functions of LAT2 protein still needs further investigation.

## 5. Conclusions

BDL increased the expression and function of LAT1 at the BBB of rats, which was attributed to increased levels of plasma bilirubin. Bilirubin upregulated LAT1 expression by activating the AhR pathway.

## Figures and Tables

**Figure 1 biomedicines-09-01320-f001:**
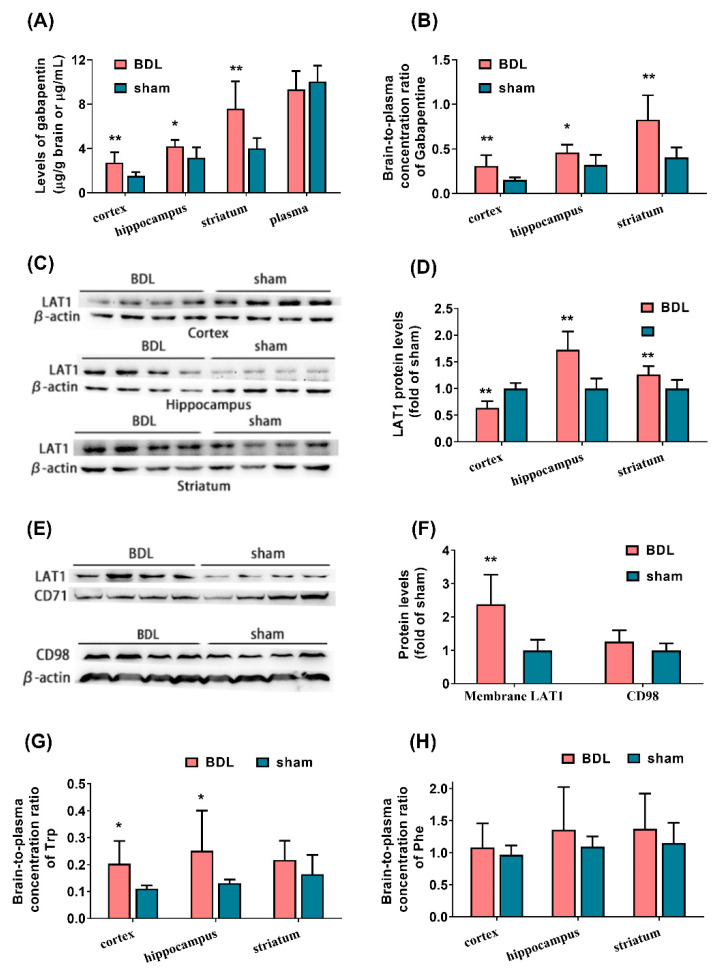
Effects of BDL on function and expression of LAT1 in brain tissues of rats. Concentrations of gabapentin in the cortex, hippocampus, striatum, and plasma of rats (**A**) and the brain-to-plasma concentration ratio of gabapentin (**B**). Levels of total LAT1 protein (**C**,**D**) in the cortex, hippocampus, and striatum of rats. Levels of CD98 and membrane LAT1 protein in cortex of rats (**E**,**F**). The brain-to-plasma concentration ratio of Trp (**G**) and Phe (**H**) in cortex, hippocampus, and striatum of rats. Data are presented as the mean ± SD (*n* = 8). * *p* < 0.05 and ** *p* < 0.01 versus sham rats.

**Figure 2 biomedicines-09-01320-f002:**
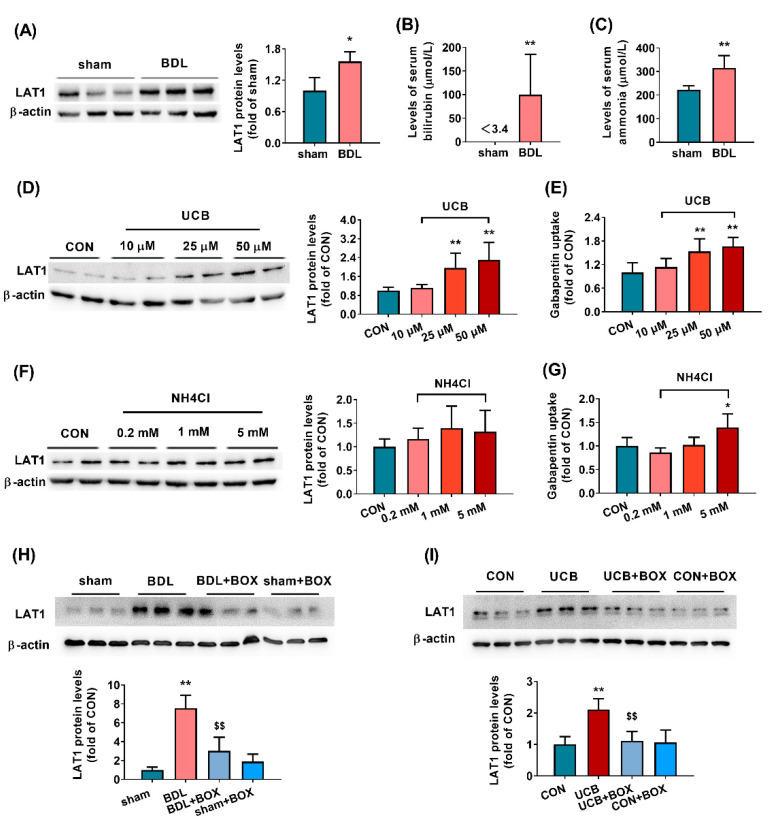
Effects of potential factors on the expression and function of LAT1 in HCMEC/D3 cells. Effects of serum from BDL rats on the expression of LAT1 (**A**). The levels of bilirubin (**B**) and ammonia (**C**) in serum of BDL and sham rats. Effects of unconjugated bilirubin (UCB) on expression (**D**) and function (uptake of gabapentin, (**E**)) of LAT1 protein. Effects of NH_4_Cl on expression (**F**) and function (uptake of gabapentin, (**G**)) of LAT1 protein. Effects of serum from BDL and sham rats in absence and presence of bilirubin oxidase (BOX) on protein levels of LAT1 in HCMEC/D3 cells (**H**). Effects of UCB and BOX on protein levels of LAT1 (**I**). Data are presented as the mean ± SD (*n* = 6). * *p* < 0.05 and ** *p* < 0.01 versus sham rats or control (CON) cells. ^$$^
*p* < 0.01 versus BDL rats or UCB treated group.

**Figure 3 biomedicines-09-01320-f003:**
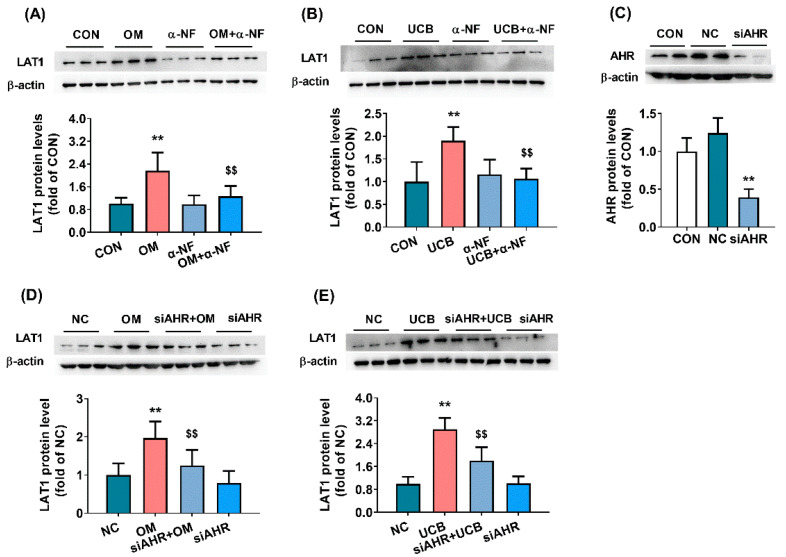
Roles of AhR in UCB-mediated regulation of LAT1 protein in HCMEC/D3 cells. Effects of AhR antagonist α-naphthoflavone (α-NF) on upregulation of LAT1 protein by omeprazole (OM, (**A**)) or UCB (**B**). The expression of the AhR protein in HCMEC/D3 cells treated with siRNA of AhR (siAHR, (**C**)). Effects of AhR gene silencing on upregulation of LAT1 protein by OM (**D**) and UCB (**E**). Data are presented as the mean ± SD (*n* = 6). ** *p* < 0.01 versus control (CON) cells or cells scrambled siRNA (NC). ^$$^
*p* < 0.01 versus cells treated with OM or UCB.

**Figure 4 biomedicines-09-01320-f004:**
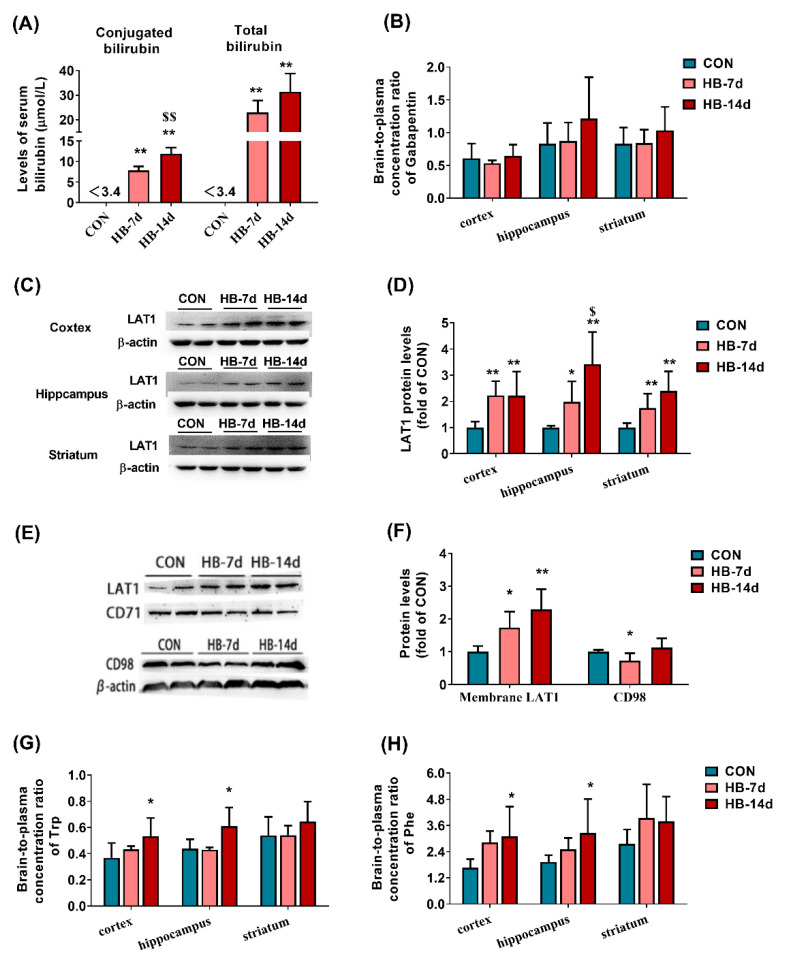
Effects of UCB treatment on LAT1 function and expression in the brain of rats. The levels of total bilirubin and conjugated bilirubin in serum (**A**) and the brain-to-plasma concentration ratio of gabapentin in the cortex, hippocampus, and striatum (**B**). Levels of total LAT1 protein (**C**,**D**) in the cortex, hippocampus, and striatum. The expression of CD98 and membrane LAT1 protein (**E**,**F**) in rat cortex. The brain-to-plasma concentration ratio of Trp (**G**) and Phe (**H**) in cortex, hippocampus and striatum. Hyperbilirubinemia rats (HB). Data are presented as the mean ± SD (*n* = 5). * *p* < 0.05 and ** *p* < 0.01 versus CON rats. ^$^
*p* < 0.05 and ^$$^
*p* < 0.01 versus HB-7d rats.

**Table 1 biomedicines-09-01320-t001:** Physiological and biochemical parameters of sham and BDL rats on the 14th day following operation.

Parameters	Sham	BDL
Body weight (BW) (g)	238.38 ± 9.34	234.88 ± 18.96
Liver weight (% BW)	3.28 ± 0.18	6.04 ± 0.91 **
Spleen weight (% BW)	0.24 ± 0.03	0.53 ± 0.11 **
ALT (IU/L)	28.23 ± 5.62	40.75 ± 9.04 **
AST (IU/L)	51.00 ± 9.70	122.15 ± 46.55 **
ALP (IU/L)	358.38 ± 84.89	681.94 ± 207.21 **
Total bilirubin (μmol/L)	<3.4	99.76 ± 80.11 **
Total Bile Acids (μmol/L)	34.25 ± 13.24	130.55 ± 73.18 **
Serum ammonia (μmol/L)	222.32 ± 16.23	314.77 ± 49.56 **

Data are presented as mean ± SD (*n* = 8). ** *p* < 0.01 vs. sham rats. BDL: bile duct ligation; ALT: alanine aminotransferase; AST: aspartate aminotransferase; ALP: alkaline phosphatase.

## Data Availability

The data presented in this study are all contained within the main body of this article.

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
