# Peer review of "Bile Duct Ligation Upregulates Expression and Function of L-Amino Acid Transporter 1 at Blood–Brain Barrier of Rats via Activation of Aryl Hydrocarbon Receptor by Bilirubin"

_biomedicines, 2021, doi:10.3390/biomedicines9101320_

Round 1

Reviewer 1 Report

The research article "Bile duct ligation upregulates expression and function of L- amino acid transporter 1 at blood-brain barrier of rats via activation of aryl hydrocarbon receptor by bilirubin" investigates mechanistic link between liver failure and increased levels of aromatic amino acids, ultimately causing hepatic encephalopathy. The authors report that the principal mechanism behind this is through the action of increase plasma bilirubin via aryl hydrocarbon receptor and consequential induction of LAT1 transporter and increased permeability of BBB for AAAs. Overall, this work is rather well conceived, structured and presented, globally the experiments support the main conclusions. However, certain issues must be addressed by the authors.
Major ones:
- The authors convincingly show that bilirubin induces LAT1, however, it is unclear whether it is the only factor. If authors still have some serum from BDL rats used in experiments for Fig. 2, it would be interesting to treat these samples with bilirubin oxidase to specifically degrade the compound and then assess residual LAT1-inducing activity.
- It is unclear why the authors did not perform treatment of the animals with AhR antagonist alpha-naphthoflavone to confirm the effect of the AhR in vivo?
- Discussion mentions that "brain-to-plasma concentration ratios of Phe and Trp in striatum of hyperbilirubinemia rats only showed a
trend to increase, indicated that there also existed other factors to affect the transport of Phe and Trp across BBB under BDL-induced liver failure status". However, this is indeed only correct for striatum; moreover, the data for BDL is presented for the whole brain and in µg/ml while for HB it is presented for brain compartments and as tissue to plasma ratio. This is confusing, the authors should present the data in  
comparable manner and make proper conclusion. Also, the materials and methods part mentions only Phe and Trp quantifications in BDL rats, was the same method used for HB ones?
- It is also impossible to clearly understand how many animals were used in every type of experiment. Details of grouping should be clearly described in material and methods section and preferably mentioned in the results section
- On the Figure 4E striatum and hippocampus are just copies of the same western blot. However, I see that they are different in the attached raw data. 
- The appendix, supplementary and raw data are provided in a strange manner. The supplementary data file is just an arranged copy of raw data files, though normally one expect to see there some additional data, such as those presented in appendix. The data in appendix are not mentioned in the text at all. I suggest that authors should leave just an archive of raw files and create a proper supplement without compilation of raw westerns and with figures A1 and A2 in it, and add their proper mentioning in the text.

Minor issues:
- overall the text contains a significant number of language errors, e.g. authors use everywhere word "expressions", so the text should be double checked. Some more important noticed typos:
    line 36: evaluates->elevates
    line 64: "orders ... are"-> order ... is
    line 139 and others: RPIM->RPMI
- the authors mark total bilirubin levels "ND" in the table 1, and only mention that it is <3.4 in the annotation. I think it would be more clear if they put <3.4 on the table straight away. Also, abbreviations ALT, AST and ALP should be explained at least once.
- abbreviations of the HB-7d and HB-14d rats should be explained in the corresponding results section, otherwise it might be difficult to understand

Author Response

Response to Reviewer 1 Comments

We have uploaded the revised manuscript entitled “Bile duct ligation upregulates expression and function of L-amino acid transporter 1 at blood-brain barrier of rats via activation of aryl hydrocarbon receptor by bilirubin”. Thank you for your kind suggestions. We have revised the manuscript according to your suggestions and highlighted the changes in our manuscript using ‘track changes’ function, meanwhile, the key parts of the revision are marked in blue in the manuscript. Below we have given an itemized list of responses for all the comments and questions you raised.

Point 1:

The authors convincingly show that bilirubin induces LAT1, however, it is unclear whether it is the only factor. If authors still have some serum from BDL rats used in experiments for Fig. 2, it would be interesting to treat these samples with bilirubin oxidase to specifically degrade the compound and then assess residual LAT1-inducing activity.

Response 1:

Thank you for your suggestions. According to your advice, we added another experiment to investigate whether serum of BDL rats treated with bilirubin oxidase still affected expressions of LAT1 protein in HCMEC/D3 cells. Our results showed that bilirubin oxidase almost abolished the up-regulation of LAT1 protein by bilirubin and serum of BDL rats, further demonstrating that upregulation of LAT1 protein by serum of BDL rats was mainly attributed to the increased bilirubin levels.

We have rewritten “Abstract”, “Materials and Methods”, “Results” and “Discussion” in the revised manuscript as follows.

In “ Abstract” 

……HCMEC/D3 served as in vitro BBB model, and the data showed that both the serum of BDL rats and bilirubin induced LAT1 expression and function, while bilirubin oxidase almost abolished the upregulation of LAT1 protein by bilirubin and serum of BDL.

In “      Materials and Methods

2.1. Reagents

……Omeprazole, and α-naphthoflavone and the bilirubin oxidase (BOX) were obtained from Aladdin Co., Ltd (Shanghai, China). ……

2.5. HCMEC/D3 culture and drug treatment

……The cultured cells were collected for determining protein expression and function of LAT1, respectively. The MTT assay showed that these agents at the concentrations tested did not damage cell viability. To further investigate whether serum of BDL rats induced the expression of LAT1 protein due to the increased bilirubin levels, LAT1 protein levels were also measured in HCMEC/D3 cells following treatment with UCB (50 μM) or serum of BDL rats in absences and presences of bilirubin oxidase (50 μg/ml). The levels of bilirubin oxidase we used were cited from previous reports [41].……

In “Results

3.3. Effects of UCB and ammonia on expressions and functions of LAT1 in HCMEC/D3 cells

  ……

50 mM UCB significantly increased the protein expression and function (gabapentin uptake) of LAT1 to 2.3 fold and 1.7 fold of control cells (Figure 2D, E), respectively. 5 mM NH4Cl only showed little enhancement on uptake of gabapentin (Figure 2F, G). The MTT assay showed that these agents at the concentrations tested did not damage cell viability (Figure A3). Further study showed that the treatment of bilirubin oxidase almost abolished the upregulation of LAT1 protein by UCB and serum of rats (Figure 2H and I), demonstrating that upregulation of LAT1 protein by serum of BDL rats was mainly attributed to the increased bilirubin levels.

Figure 2H and I. Effects of serum from BDL and sham rats with or without bilirubin oxidase (BOX) treatment on protein levels of LAT1 in HCMEC/D3 cells (H). Effects of unconjugated bilirubin (UCB) and BOX on protein levels of LAT1 (I). Data are presented as the mean ± SD (n=6). *p<0.05 and **p<0.01 versus sham rats or control (CON) cells. $p<0.05 and $$p<0.01 versus BDL rats or UCB treated group (UCB).

In “Discussion

The results showed that only bilirubin concentration-dependently enhanced the function and expression of LAT1 in HCMEC/D3 cells, and BOX treatment obviously de-creased the LAT1 inducing effects of bilirubin and serum of BDL rats. Above results demonstrated that the increased bilirubin levels in serum of BDL rats are the main compound inducing LAT1 expression in BBB.

Point 2:

It is unclear why the authors did not perform treatment of the animals with AhR antagonist alpha-naphthoflavone to confirm the effect of the AhR in vivo?

Response 2:

We are sorry not to clearly illustrate it. In this research, we aimed to investigate whether BDL-induced liver failure increased levels of brain AAAs via upregulating expression of LAT1 on BBB, and to investigate the mechanism that BDL-induced liver failure induced expression of LAT1 protein. Data from HCMEC/D3 cells showed that abnormally increased bilirubin levels are the main factors that BDL induced expression of brain LAT1 protein. The deduction was further confirmed using hyperbilirubinemia rats. In vitro data demonstrated that bilirubin upregulated expression of LAT1 protein via activating AhR, which were attenuated by both AhR antagonist and AhR silencing.

Bilirubin is considered to be a ligand of AhR [1,2] and relationships of AhR and LAT1 expression as well as bilirubin have been widely investigated [3-7]. For example, bilirubin was reported to induce expression of UGT1A1 [3] and CYP1a1 [4] via activating the AhR pathway. Mouse experiment also showed that the immunomodulatory properties of UCB are mediated through an AhR-dependent pathway [5]. Activation of AhR has been demonstrated upregulation expression of LAT1 protein in MCF-7, MDA-MB-23 cells [6] and human bronchial epithelial cells[7]. Above results could support our conclusion that bilirubin activates AhR to upregulate LAT1 expression. Therefore, whether AhR antagonist reversed upregulation of LAT1 protein by bilirubin in rats was not performed in the study. On the other hand, AhR plays many important roles in maintaining homeostasis in the whole body [8]–[10], the use of pan-antagonist of it may induce disorders of body metabolism balance and influence the results of experiment, this also is one of concern we didn’t perform treatment of the animals with AhR pan-antagonist.

Citations:

[1]      Shinde, R. & McGaha, T. L. The Aryl Hydrocarbon Receptor: Connecting Immunity to the Microenvironment. Trends Immunol. 39, 1005–1020 (2018).

[2]   Bock KW, Köhle C. The mammalian aryl hydrocarbon (Ah) receptor: from mediator of dioxin toxicity toward physiological functions in skin and liver. Biol Chem. 2009 Dec;390(12):1225-3

[3]      Togawa, H.; Shinkai, S.; Mizutani, T. Induction of Human UGT1A1 by Bilirubin through AhR Dependent Pathway. Drug Metab. Lett. 2008, 2 (4), 231–237. https://doi.org/10.2174/187231208786734120.

[4]      Sinal, C. J.; Bend, J. R. Aryl Hydrocarbon Receptor-Dependent Induction of Cyp1a1 by Bilirubin in Mouse Hepatoma Hepa 1c1c7 Cells. Mol. Pharmacol. 1997, 52 (4), 590–599. https://doi.org/10.1124/mol.52.4.590.

[5]   Longhi MS, Vuerich M, Kalbasi A. et al. Bilirubin suppresses Th17 immunity in colitis by upregulating CD39. JCI Insight. 2017 May 4;2(9):e92791. https://doi.org/ 10.1172/jci.insight.92791

[6]   Tomblin, J. K. et al. Aryl hydrocarbon receptor (AHR) regulation of L-Type Amino Acid Transporter 1 (LAT-1) expression in MCF-7 and MDA-MB-231 breast cancer cells. Biochem. Pharmacol. 106, 94–103 (2016).

[7]   Le Vee, M., Jouan, E., Lecureur, V. & Fardel, O. Aryl hydrocarbon receptor-dependent up-regulation of the heterodimeric amino acid transporter LAT1 (SLC7A5)/CD98hc (SLC3A2) by diesel exhaust particle extract in human bronchial epithelial cells. Toxicol. Appl. Pharmacol. 290, 74–85 (2016).

[8]      Hao, N. & Whitelaw, M. L. The emerging roles of AhR in physiology and immunity. Biochem. Pharmacol. 86, 561–570 (2013).

[9]      Shinde, R. & McGaha, T. L. The Aryl Hydrocarbon Receptor: Connecting Immunity to the Microenvironment. Trends Immunol. 39, 1005–1020 (2018).

[10]    Murray, I. A., Patterson, A. D. & Perdew, G. H. Aryl hydrocarbon receptor ligands in cancer: friend and foe. Nat. Rev. Cancer 14, 801–814 (2014).

And according your suggestion, we have rewritten the “Discussion” in the revised manuscript as follows.

  Bilirubin is considered to be an endogenous ligand of AhR [44,45], and Several studies have demonstrated that bilirubin induces the expression of UGT1A1 [42] and CYP1a1 [43] via activating the AhR pathway. Mouse experiment also showed that UCB has immunomodulatory effects, which also was mediated by an AhR-dependent pathway [46]. Activation of AhR has been also demonstrated to upregulate the expression of LAT1 protein in MCF-7, MDA-MB-23 cells [47], and in human bronchial epithelial cells [48]. We found that omeprazole, an AhR agonist, also could induce the expression of LAT1 protein in HCMEC/D3 cells, demonstrating that LAT1 may be a target of AhR.

Point 3:

Discussion mentions that "brain-to-plasma concentration ratios of Phe and Trp in striatum of hyperbilirubinemia rats only showed a trend to increase, indicated that there also existed other factors to affect the transport of Phe and Trp across BBB under BDL-induced liver failure status". However, this is indeed only correct for striatum; moreover, the data for BDL is presented for the whole brain and in µg/ml while for HB it is presented for brain compartments and as tissue to plasma ratio. This is confusing, the authors should present the data in comparable manner and make proper conclusion. Also, the materials and methods part mentions only Phe and Trp quantifications in BDL rats, was the same method used for HB ones?

Point 3 A:

Discussion mentions that "brain-to-plasma concentration ratios of Phe and Trp in striatum of hyperbilirubinemia rats only showed a trend to increase, indicated that there also existed other factors to affect the transport of Phe and Trp across BBB under BDL-induced liver failure status". However, this is indeed only correct for striatum;

Response 3A:

Thanks for your advice. We removed the inaccurate conclusion in the “Discussion” and rewritten it in the revised manuscript as follows.

Moreover, LAT2 also mediates transport of Phe and Trp [49], and LAT1 and LAT2 functions are complementary [50]. Whether hyperbilirubinemia and BDL also alter expression and functions of LAT2 protein still needs further investigation.

Citations:

  1. Albrecht J, Zielińska M. Exchange-mode glutamine transport across CNS cell membranes. Neuropharmacology. 2019 Dec 15;161:107560.
  2. Errasti-Murugarren E, Palacín M. Heteromeric Amino Acid Transporters in Brain: from Physiology to Pathology. Neurochem Res. 2021 Feb 19. doi: 10.1007/s11064-021-03261

Point 3B. moreover, the data for BDL is presented for the whole brain and in µg/ml while for HB it is presented for brain compartments and as tissue to plasma ratio.  

Response 3B:

 We are sorry not to clearly illustrate it. According to your suggestion, we have rewritten the “Results” and rearranged Figure 1, Figure 4, Figure A2, A4 and A5 in the revised manuscript as follows:

3.2. BDL increases the function and expression of LAT1 in brain tissues of rats

……

The concentrations of Trp and Phe in brain tissues of BDL rats were also measured and the brain-to-plasma concentration ratios were calculated. It was in line with the increased function of brain LAT1 that BDL significantly increased the levels of Trp and Phe in cortex, hippocampus and striatum of rats (Figure A2), and obviously elevated the ratio of brain-plasma of Trp in cortex and hippocampus of rats (Figure 1G).……

3.5. Effects of hyperbilirubinemia on brain LAT1 expression and function

……

The brain distributions of gabapentin in HB-7d rats and HB-14d rats were further measured. The results showed that the effects of hyperbilirubinemia on brain distributions of gabapentin were time-dependent. Compare with control rats, the brain of HB-14d rats exhibited higher concentrations of gabapentin, and the significance was found in the hippocampus and striatum of HB-14 rats (Figure A4). The ratio of brain-plasma of gabapentin also showed a trend to increase in hippocampus and striatum of HB-14d rats, although no significance was obtained (Figure 4B).

……

Concentrations of Phe and Trp in the brain of hyperbilirubinemia rats were also measured. The results showed that Phe concentration significantly increased in the cortex of HB-14d rats, while Trp concentration significantly increased in the cortex of HB-7d rats (Figure A5). The brain-to-plasma concentrations ratio was also calculated, and the data showed that hyperbilirubinemia significantly increased the brain-to-plasma concentration ratio of Phe and Trp in the cortex and hippocampus of HB-14d rats, which possibly results from UCB-induced LAT1 upregulation (Figure 4G, H).

Point 3C: Also, the materials and methods part mentions only Phe and Trp quantifications in BDL rats, was the same method used for HB ones?

Response 3C.

We are sorry not to clearly illustrate it. According to your suggestion, we have rewritten   “Materials and Methods” in the revised manuscript as follows.

In “Materials and Methods

2.8. Effects of hyperbilirubinemia on LAT1 function and expression in brain of rats

……Then, the rats were sacrificed under light diethyl ether at 40 min following dose of gabapentin. Plasma and brain samples were obtained, then the levels of gabapentin, Phe, and Trp in the brain and plasma as well as expression of LAT1 in the brain were measured by using the same methods described above.

Point 4:

It is also impossible to clearly understand how many animals were used in every type of experiment. Details of grouping should be clearly described in material and methods section and preferably mentioned in the results section.

Response 4:

Thank you for your suggestions and we apologize for not clearly illustrate it. We have rewritten and clearly described it in the “materials and methods” and “legends” section of the revised manuscript.

In “materials and methods

2.3. Development of BDL rats

BDL rats were developed according to method previously described [36]. Sham-operated (sham) rats received the same experimental procedure without bile duct ligation. On day 14 following operation, 8 BDL rats and 8 sham rats were used for the following experiments.……

2.8. Effects of hyperbilirubinemia on LAT1 function and expression in brain of rats

To further study effects of UCB on the function and expression of LAT1 in BBB, hyper-bilirubinemia rats were induced by intraperitoneal dose of UCB (85.5 μmol/kg, once) for 7 days (HB-7d) and 14 days (HB-14d), respectively, according to the method previously described [37,42]. The control rats only received vehicle, and every group have 5 rats.……

Point 5:

On the Figure 4E striatum and hippocampus are just copies of the same western blot. However, I see that they are different in the attached raw data.

Response 5:

Thank you for pointing out this mistake, and we are very sorry for our negligence. As you see, the raw data is right, we corrected the Figure 4C in the revised manuscript as follows:

Figure 4C and D. Levels of total LAT1 protein (C, D) in the cortex, hippocampus and striatum.

Thanks again for your careful and responsibility.

Point 6:

The appendix, supplementary and raw data are provided in a strange manner. The supplementary data file is just an arranged copy of raw data files, though normally one expect to see there some additional data, such as those presented in appendix. The data in appendix are not mentioned in the text at all. I suggest that authors should leave just an archive of raw files and create a proper supplement without compilation of raw westerns and with figures A1 and A2 in it, and add their proper mentioning in the text.

Response 6:

Thanks for your suggestion. We have corrected them in the revised manuscript.

Point 7 minor issues

Overall, the text contains a significant number of language errors, e.g. authors use everywhere word "expressions", so the text should be double checked. Some more important noticed typos:
    line 36: evaluates->elevates
    line 64: "orders ... are"-> order ... is
    line 139 and others: RPIM->RPMI
- the authors mark total bilirubin levels "ND" in the table 1, and only mention that it is <3.4 in the annotation. I think it would be more clear if they put <3.4 on the table straight away. Also, abbreviations ALT, AST and ALP should be explained at least once.

- abbreviations of the HB-7d and HB-14d rats should be explained in the corresponding results section, otherwise it might be difficult to understand.

Response 7

Thanks very much for your kind suggestions, we apologize for our negligence. According to your advice, we double checked the abbreviations, grammar, phrasing and punctuation, and corrected all above errors using the “track function” in the revised manuscript.

Reviewer 2 Report

Overall a well written scientific paper focusing on the role of LAT1 function in hepatic encephalopathy.

Some minor modifications are needed for better clarity:

Line 26 - Several studies [3–13] have demonstrated that liver failure significantly [36] evaluates (don’t’ understand what this means) concentrations of aromatic amino acids (AAAs) in brain or cerebrospinal fluid, 37 including phenylalanine (Phe), tyrosine (Tyr) and tryptophan (Trp), whose increases are 38 often positively related to grades of hepatic encephalopathy [3,4,6].

Line 324 - However, it was contrast to (please reformulate to be more clear) the increased function 324 of cortex LAT1 that BDL significantly decreased expression of total LAT1 protein in cortex 325 of rats.

Author Response

Response to Reviewer 2 Comments

We have uploaded the revised manuscript entitled “Bile duct ligation upregulates expression and function of L-amino acid transporter 1 at blood-brain barrier of rats via activation of aryl hydrocarbon receptor by bilirubin”. Thank you for your kind suggestions. We have revised the manuscript according to your suggestions and highlighted the changes in our manuscript using ‘track changes’ function, meanwhile, the key parts of the revision are marked in blue in the manuscript. Below we have given an itemized list of responses for all the comments and questions you raised.

Point 1:

Overall a well written scientific paper focusing on the role of LAT1 function in hepatic encephalopathy.

Some minor modifications are needed for better clarity:

Line 26 - Several studies [3–13] have demonstrated that liver failure significantly [36] evaluates (don’t’ understand what this means) concentrations of aromatic amino acids (AAAs) in brain or cerebrospinal fluid, including phenylalanine (Phe), tyrosine (Tyr) and tryptophan (Trp), whose increases are often positively related to grades of hepatic encephalopathy [3,4,6].

Line 324 - However, it was contrast to (please reformulate to be more clear) the increased function of cortex LAT1 that BDL significantly decreased expression of total LAT1 protein in cortex of rats.

Response 1:

Thanks a lot for your kind suggestions to our manuscript. we apologize for our negligence. According your advice, we double checked the abbreviations, grammar, phrasing and punctuation, and corrected all above errors using the “track function” as follows and in the revised manuscript.

In “Introduction

Several studies [3–13] have demonstrated that liver failure significantly elevates con-centrations of aromatic amino acids (AAAs) in the brain or cerebrospinal fluid, including phenylalanine (Phe), tyrosine (Tyr), and tryptophan (Trp), whose increases are often positively related to grades of hepatic encephalopathy [3,4,6].

In “Disscussion

However, it was in contrast to the increased concentration of gabapentin in the cortex (indicated the upregulation of LAT1 functions) that BDL significantly decreased the expression of total LAT1 protein in the cortex of rats.……

In addition, we also added some experiments to further prove the inducing-effect of bilirubin on LAT1 expression in the manuscript as follows:

In “ Abstract” 

……HCMEC/D3 served as in vitro BBB model, and the data showed that both the serum of BDL rats and bilirubin induced LAT1 expression and function, while bilirubin oxidase almost abolished the upregulation of LAT1 protein by bilirubin and serum of BDL.

In “      Materials and Methods

2.1. Reagents

……Omeprazole, and α-naphthoflavone and the bilirubin oxidase (BOX) were obtained from Aladdin Co., Ltd (Shanghai, China). ……

2.5. HCMEC/D3 culture and drug treatment

……The cultured cells were collected for determining protein expression and function of LAT1, respectively. The MTT assay showed that these agents at the concentrations tested did not damage cell viability. To further investigate whether serum of BDL rats induced the expression of LAT1 protein due to the increased bilirubin levels, LAT1 protein levels were also measured in HCMEC/D3 cells following treatment with UCB (50 μM) or serum of BDL rats in absences and presences of bilirubin oxidase (50 μg/ml). The levels of bilirubin oxidase we used were cited from previous reports [41].……

In “Results

3.3. Effects of UCB and ammonia on expressions and functions of LAT1 in HCMEC/D3 cells

  ……

50 mM UCB significantly increased the protein expression and function (gabapentin uptake) of LAT1 to 2.3 fold and 1.7 fold of control cells (Figure 2D, E), respectively. 5 mM NH4Cl only showed little enhancement on uptake of gabapentin (Figure 2F, G). The MTT assay showed that these agents at the concentrations tested did not damage cell viability (Figure A3). Further study showed that the treatment of bilirubin oxidase almost abolished the upregulation of LAT1 protein by UCB and serum of rats (Figure 2H and I), demonstrating that upregulation of LAT1 protein by serum of BDL rats was mainly attributed to the increased bilirubin levels.

Figure 2H and I. Effects of serum from BDL and sham rats with or without bilirubin oxidase (BOX) treatment on protein levels of LAT1 in HCMEC/D3 cells (H). Effects of unconjugated bilirubin (UCB) and BOX on protein levels of LAT1 (I). Data are presented as the mean ± SD (n=6). *p<0.05 and **p<0.01 versus sham rats or control (CON) cells. $p<0.05 and $$p<0.01 versus BDL rats or UCB treated group (UCB).

In “Discussion

The results showed that only bilirubin concentration-dependently enhanced the function and expression of LAT1 in HCMEC/D3 cells, and BOX treatment obviously de-creased the LAT1 inducing effects of bilirubin and serum of BDL rats. Above results demonstrated that the increased bilirubin levels in serum of BDL rats are the main compound inducing LAT1 expression in BBB.

Round 2

Reviewer 1 Report

The authors have addressed the raised issues thoroughly and appropriately, clearly enhancing the quality of the work. In the current stat it surely can be published. 

Reviewer 2 Report

No comments, all modifications have been made.